# Mechanical and Static Stab Resistant Properties of Hybrid-Fabric Fibrous Planks: Manufacturing Process of Nonwoven Fabrics Made of Recycled Fibers

**DOI:** 10.3390/polym11071140

**Published:** 2019-07-03

**Authors:** Yu-Chun Chuang, Limin Bao, Mei-Chen Lin, Ching-Wen Lou, TingAn Lin

**Affiliations:** 1Interdisciplinary Graduate School of Science and Technology, Shinshu University, Nagano Prefecture 390-8621, Japan; 2Department of Science and Technology, Graduate School of Medicine, Science and Technology, Shinshu University, Nagano Prefecture 390-8621, Japan; 3Department of Medical Research, China Medical University Hospital, China Medical University, Taichung City 40402, Taiwan; 4Department of Bioinformatics and Medical Engineering, Asia University, Taichung City 41354, Taiwan; 5Innovation Platform of Intelligent and Energy-Saving Textiles, School of Textiles, Tianjin Polytechnic University, Tianjin 300387, China; 6College of Textile and Clothing, Qingdao University, Qingdao 266071, China; 7Department of Chemical Engineering and Materials, Ocean College, Minjiang University, Fuzhou 350108, Fujian, China

**Keywords:** hybrid-fabric fibrous plank, recycled selvages, stab resistance

## Abstract

With the development of technology, fibers and textiles are no longer exclusive for the use of clothing and decoration. Protective products made of high-strength and high-modulus fibers have been commonly used in different fields. When exceeding the service life, the protective products also need to be replaced. This study proposes a highly efficient recycling and manufacturing design to create more added values for the waste materials. With a premise of minimized damage to fibers, the recycled selvage made of high strength PET fibers are reclaimed to yield high performance staple fibers at a low production cost. A large amount of recycled fibers are made into matrices with an attempt to decrease the consumption of new materials, while the combination of diverse plain woven fabrics reinforces hybrid-fabric fibrous planks. First, with the aid of machines, recycled high strength PET fibers are processed into staple fibers. Using a nonwoven process, low melting point polyester (LMPET) fibers and PET staple fibers are made into PET matrices. Next, the matrices and different woven fabrics are combined in order to form hybrid-fabric fibrous planks. The test results indicate that both of the PET matrices and fibrous planks have good mechanical properties. In particular, the fibrous planks yield diverse stab resistances from nonwoven and woven fabrics, and thus have greater stab performance.

## 1. Introduction

In recent years, fibrous and textile products are applied to many other fields than garments and decoration. Due to the flexibility, light weight, and easy process, they can be easily combined with other materials. In addition to the intrinsic properties, they can also yield other required properties via the employment of different organizational structures and manufacturing processes. Hence, fibrous and textile materials have a great diversity of applications, from domestic and industrial purposes to custom-made special functionality requests. The amounts of selvages and textile wastes soar as a result of the manufacturers producing increasingly demanded protective textiles, including some high price, high strength, and high modulus fibrous materials. In the production of weaving fabrics, due to the limitations of the machine and the quality of the product, the width of the fabric usually produced must be slightly larger than the actual required width. Therefore, almost all fabrics need to be cut and then sold. A large amount of discarded cloth selvages are produced in the process. On the other hand, these protective products will be phased out after the end of their useful life. However, the high modulus fibers used in these protective products may only be slightly damaged. It will be a pity to just discard or burn these materials or use them as fillers [1], especially the materials that are left after the production of protective products [2,3]. Moreover, protective products with expired service life may be discarded when the constituent high modulus fibers are slightly damaged. Hence, a highly efficient recycling and reclaiming design can process the materials to have new additive values.

Recycling helps to reduce the demands for source and energy [1,4,5]. It also involves many developed techniques, the majority of which is about fibers and not commercialized yet. The major recycling methods for functional fibers or composites are mechanical recycling, heat recovery, and chemical recovery [4,5]. Heat recovery employs combustion which is able to separate resin and fibers, both of which have greatly different pyrolysis temperature. However, due to possible damage to fiber, heat recovery also requires the facility to generate and store the waste gas and liquid. When wrongly operated, heat recovery may cause considerable carbon dioxide and toxic substance that increases environmental burden [6,7,8]. By contrast, chemical recovery easily damages fibers and generates complex byproducts, which adds difficulty to the recycling. Additionally, machine crushing is suitable for recycling fibers with different lengths. It is easily operated and thus the most commonly used. In brief, recycling and reclaiming techniques and related studies engage increasingly more attention [9,10,11].

Therefore, this study aims to recycle and reclaim high strength PET waste fibers with minimum possible damage in order to regain comparatively cheaper high-performance fibers. Instead of using newly produced fibers, a great number of recycled fibers are made into nonwoven fabrics, which are then combined with different reinforcing woven fabrics, thereby producing stab resistant hybrid-fabric fibrous planks. However, we believe that an efficient recycling process and design can re-add new value to the material. There are many studies indicating that the stab resistance of protective items is created by the friction force between fabrics and yarns [12,13,14,15]. For example, fabrics that are immersed in a shear thickening fluid (STF) exhibit better stab resistance because the presence of STF particles strengthens the friction force between the fabrics and yarns [16,17,18]. Similarly, friction force between the fabrics and yarns can also be obtained when the fabrics are made with a higher fabric density, immersed in resin, or stabilized with rubber threads [2,19]. Regardless of the stab resistant materials, the stab resistance performance is primarily dependent on the friction of the materials against the impact objects. This study aims to propose an efficient method to recycle the waste woven selvages and reuse these fibers to develop a flexible stab resistance hybrid fabric composites that may be used in the protective clothing field and geotextiles field. Rather than the traditional rigid protective materials, this study designs flexible protective materials with a multi-layered fabric structure. The combination of nonwoven and woven fabrics provides hybrid planks with highly improved stab resistance. The resulted flexible hybrid-fabric fibrous planks do not render as much burden to the human body as the traditional stiff composite protective items [20,21], and can be easily made into diverse products or serve as a reinforcing item in any required secondary process. In addition, we used a nonwoven process in this study. The process technology has the advantages of fast, low cost and high output, and can fully mix two or more kinds of fiber materials and composite multi-layer fabrics to make a flexible stab-resistant hybrid fabric composite material designed by us.

## 2. Materials and Methods

### 2.1. Materials

Recycled high strength polyester (PET) selvages (Chien Chen Textile, city, Taiwan) have a fiber fineness of 1000D/192f, fiber length of 40–65 mm, and single fiber strength of 8g/d (Figure 1). Both carbon and aramid plain woven fabrics were purchased from Jinsor-Tech Industrial Co., Taichung City, Taiwan and the physical properties are listed in Table 1. Basalt woven fabrics (Yurak International, Taichung City, Taiwan) are composed of basalt fiber bundles at both warp and weft directions with a fineness of 2970 D and an areal density of 328 g/m^2^ (as shown in Table 1 and Figure 2). Low-melting-point PET (LMPET) staple fibers (Far Eastern New Century, Taiwan) have a fineness of 4 D and a length of 51 mm, and are composed of a skin–core structure. The melting points of the skin and core are 110 °C and 265 °C.

### 2.2. Method

The principal material in this study is recycled high strength PET selvages. In general, the staple fibers that are used in non-woven fabrics are wavy or crimp in order to increase the friction. However, the waste selvages of woven fabric are usually cut from the edge of woven fabric made by the filament or continue yarn and still with woven fabric structure. Thus, we had to break, dispersion, recycled and then reused these fibers. Therefore, the high strength PET waste selvage were processed with the opening into recycled PET staple fibers. The PET staple fibers were mixed with low-melting point polyester (LMPET) fibers at ratios of 9:1, 7:3, and 5:5 to form high strength PET matrices by a needle punching machine (needle punching machine, SNP120SH6, Shoou Shyng Machinery Co., Ltd., New Taipei City, Taiwan) with a needle-punched speed of 200 needles/min and a line speed of 2.3 m/min. During the needle-punching process, the needles were pressed in from the direction of the vertical fabric surface to laminate and bonded the multilayer web or multilayer fabric together. Next, pure LMPET fibers were also made into LMPET layers by the nonwoven process, which serves as the adhesive layer between the PET matrix and reinforcing the woven fabric. Different reinforcing woven fabrics were used, including basalt, carbon-fiber, and aramid plain woven fabrics. The sandwich-structured laminates were hot pressed into the hybrid-fabric fibrous planks, which were treated at 130 °C at a speed of 0.2 m/min and hot pressure with 10 MPa (two-wheel hot press machine, CW-NEB, Chiefwell Engineering Co., Ltd., New Taipei City, Taiwan). The manufacturing process was denoted in Figure 3 and Table 2. Finally, the air permeability, tensile strength, and tearing strength, bursting strength, and static stab resistance of hybrid-fabric fibrous planks were tested to examine the influence of content of recycled high strength PET fibers and the employment of hot pressing.

### 2.3. Tests

#### 2.3.1. Air Permeability

The air permeability of hybrid-fabric fibrous planks was measured using an air permeability tester (TEXTEST FX3300) as specified in ASTM D737 (Standard Test Method for Air Permeability of Textile Fabrics). Sample size is 25 cm × 25 cm. Ten samples for each specification were used in order to have the mean.

#### 2.3.2. Tensile Strength

As specified in ASTM D5035, the tensile strength of hybrid-fabric fibrous planks was measured at a cross head tensile speed of 300 mm/min using an Instron 5566 (Instron, Norwood, MA, USA). The distance between a pair of pneumatic clamps is 75 mm. Six samples for each specification along the cross machine direction (CD) and machine direction (MD) were used. Samples have a size of 25.4 mm × 180 mm.

#### 2.3.3. Tearing Strength

The tearing strength of hybrid-fabric fibrous planks was measured as specified in ASTM D5587. Samples were prepared according to the trapezoid method and had two equal altitudes and two parallel bases of 75 mm and 150 mm. The short base had a perpendicular cut with a length of 15 mm in the center. The distance between two clamps is 25 mm and the test rate is 300 mm. Six samples for each specification along the CD and MD were taken in order to have the mean.

#### 2.3.4. Bursting Strength

As specified in ASTM D3787, a universal tester (Instron 5566, Instron, Norwood, MA, USA) that is equipped with a 25.4-mm-diameter hemispherical probe was used to measure the bursting strength of hybrid-fabric fibrous planks at a rate of 100 mm/min. Six samples (150 mm × 150 mm) for each specification were used and the maximum bursting strength was recorded.

#### 2.3.5. Static-Stab Resistance Test

The static puncture resistance of samples was measured at a puncture rate of 508 mm/min using a universal strength testing machine (Instron5566, Norwood, MA, USA) as specified in ASTM F1342. Samples have a size of 100 mm × 100 mm. The diameter of the puncture probe is 4.5 mm. Six samples for each specification were used for the test in order to have the average static puncture resistance, standard deviation, and coefficient of variation (as shown in Figure 4).

## 3. Results

### 3.1. Mechanical Property of Recycle High Strength PET Matrices

Table 3 shows the mechanical property of recycled high strength PET matrices, including tensile strength, tearing strength, and air permeability. The mechanical properties are discussed based on the employment of hot pressing, the content of recycled PET fibers, and fiber orientation (i.e., the direction that the majority of fibers are aligned). Except for the tensile load, the employment of hot pressing significantly influences the elongation, tearing strength, tearing elongation, and air permeability. The LMPET fibers are melted to form thermal bonding points in the high strength PET matrices as a result of hot pressing. The thermal bonding points primarily stabilize the fabric structure and restrain the slip of fibers, which is proven by the results of tensile elongation of the matrices [22,23]. However, the thermal bonding points have a lower strength than single fiber strength of high strength PET fibers, and employment of hot pressing can hardly affect the tensile strength of the matrices.

As for the tearing strength test, the matrices have a large area, which enable thermal bonding points to resist the tearing force as well as restrain the slip of fibers. Therefore, hot pressing has a positive influence on the tearing strength and elongation of the matrices. In particular, when composed of more recycled high strength PET fibers, the matrices exhibit greater tensile and tearing strengths. Because PET fibers are gathered from PET woven selvages, they are less crimped. Subsequently, the nonwoven fabrics (i.e., PET matrices) have a low porosity and compact structure, which is proven by the low air permeability. The employment of hot pressing creates a great amount of thermal bonding points and decreases the thickness of the matrices. Hence, the matrices have low air permeability due to the high fabric density and low porosity. Moreover, when composed of 50 wt% or 70 wt% of recycled high strength PET fibers, the matrices exhibit similar tensile and tearing performances. The results are ascribed to the fact that PET fibers undermine the synergistic effect with highly crimped LMPET fibers. Comparatively, hot pressed PET matrices exhibit greater mechanical properties, and are thus used for following discussions.

### 3.2. Tensile Strength of Hybrid-Fabric Fibrous Planks

Figure 5 shows the mechanical properties of high strength PET matrices as related to the fiber blending ratios. When the content of LMPET fibers is lower than 50 wt %, the resulted PET matrices exhibit greater tensile strength. Furthermore, three plain woven fabrics are separately combined with the PET matrices for reinforcement. The hybrid-fabric fibrous planks containing basalt woven fabrics and carbon-fiber woven fabrics have comparable mechanical properties. Basalt fibers and carbon fibers have similar properties, and both fibers are fragile and cannot be bent (cf. Figure 5). Hence, when using a basalt or carbon-fiber plain woven fabric as the reinforcement, the fibrous planks exhibit similar trends in tensile strength tests, and the maximum tensile strength occurs when the fibrous planks are composed of 70 wt% of PET fibers. Specifically, the planks composed of aramid woven fabrics outperform planks composed of basalt or carbon-fiber woven fabrics in terms of tensile strength. Figure 6 shows the fractured images of different woven fabrics. Furthermore, Figure 7 indicates that the fibrous planks composed of aramid woven fabrics exhibit the highest tensile elongation due to the greater elongation rate of aramid fibers. As a result, the fibrous planks composed of aramid woven fabrics do not have a sudden decrease in the tensile strength when the aramid woven fabric is damaged. However, the high modulus fibers are commonly coated with oiling agent during the spinning and weaving process, which hampers the melted LMPET fibers to adhere. Therefore, the thermal bonding effect of LMPET fibers is insignificant, and aramid fibers slip to a greater extent. The test results show that HP9K that is composed of a lower content of LMPET fibers has the maximum tensile strength.

### 3.3. Tearing Strength of Hybrid-Fabric Fibrous Planks

Figure 8 shows the tearing strength and elongation of hybrid-fabric fibrous planks as related to fiber blending ratios. Samples are prepared with a perpendicular cut in the center beforehand. The test results show that hybrid-fabric fibrous planks consisting of aramid woven fabrics have the maximum tearing strength. Figure 9 shows that fibrous planks that are composed of greater LMPET and not hot pressed exhibit low tearing elongation, which suggests that the employment of hot press creates thermal bonding points, preventing the slip of fibers and stabilizing the structure. By contrast, both basalt and carbon-fiber woven fabrics have fragile fibers. The breakage of basalt or carbon-fiber woven fabrics causes a sudden decrease in the tearing properties, which in turn stops the test immediately. In particular, consisting of 10 wt % of LMPET fibers and 90 wt % of recycled high strength PET fibers, HP9K exhibits the highest tearing strength. The high strength PET fibers are repeatedly processed with combing and scattering to form staple fibers, and then made into nonwoven fabrics. Based on the test results, the tensile and tearing strength of the PET fibers are comparable to those of the embossing fibers, which indicates that the recycling has effective value [24,25,26].

### 3.4. Bursting Strength of Hybrid-Fabric Fibrous Planks

Figure 10 shows the bursting strength of hybrid-fabric fibrous planks as related to the fiber blending ratios. The fibrous planks that consist of a higher content of recycled PET fibers have higher bursting strength. The recycled PET fibers undergo the combing and carding processes repeatedly to form the staple fibers for nonwoven fabrics. Unlike the commonly used staple fibers in nonwoven fabrics, the recycled PET fibers are not crimped or embossing. Usually, crimped or embossing fibers contribute relatively higher friction and a uniform distribution to the nonwoven fabrics. In addition, the recycled PET fibers are chopped from complete bundles, which may possibly leave more filling and oiling agent that restrain the subsequent combing and opening processes. Based on the bursting strength, all hybrid-fabric fibrous planks have comparable bursting strength and coefficient of variation. Only when containing uneven fiber distribution or obvious fiber packing do the planks exhibit uneven and diversity in the bursting strength. Moreover, the fibrous planks consisting of a greater amount of recycled PET fibers demonstrate higher bursting strength, which suggests that the fibers are evenly distributed and the recycling PET fibers for the production of nonwoven fabrics is proven effective. The test results show that HP9K that is composed of a lower content of LMPET fibers has the maximum bursting strength about 146.8% better than the sample of HP9.

### 3.5. Static Stab-Resistance of Hybrid-Fabric Fibrous Planks

Figure 11 shows the static stab resistance of hybrid-fabric fibrous planks as related to the fiber blending ratios. Consisting of 90 wt % of recycled PET fibers in the PET matrix and Kevlar woven fabric as reinforcement, P9K demonstrates the maximum static stab resistance. This test uses a pointed probe with a diameter of 4.5 mm, and the stab resistance mechanism of fibrous planks is via the resistance against the tip of the probe as well as the frictional resistance of fibers against the pointed probe. The displacement of the fibrous planks is relatively smaller in Figure 12. The hybrid-fabric fibrous planks to resist the pointed probe via the compact plain structure of woven fabric as well as the melting status and adhesion effects caused by a great amount of LMPET fibers. The specified design of the fibrous planks is to stop the slip of fibers and increase the friction between fibers and the probe effectively. However, the recycled high modulus PET fibers are coated with oiling agent, which prevents LMPET fibers to form an adhesive layer during the hot pressing, and thus the interface bonding strength is low. Furthermore, the fibrous planks exhibit highest static stab resistance when consisting of the greatest amount of recycled PET fibers and the smallest amount of LMPET fibers about 212.6 N. Based on the test results, the static stab-resistance of the hybrid-fabric fibrous planks are comparable to those of the embossing fibers, which indicates that the recycling has effective value [24,27]. The future studies need to remove the filling and oiling agent before conducting the test for further discussion.

## 4. Conclusions

This study proposes flexible fabric-based protective planks, which are recycled high strength PET fibers by processed with minimum damage for secondary production, thereby obtaining recycled high performance fibers with relatively lower production cost. In this study, the different reinforcing woven fabrics are combined with matrices to form hybrid-fabric fibrous planks. Despite multiple combining and carding processes, the recycled PET staple fibers are proven to provide the fibrous planks with high tensile and tearing strengths. The test results indicate that recycled PET fibers remain high strength and can be made into protective products.

The test results indicate that recycled PET fibers remain high strength and can be made into protective products. The sample consisting of 10 wt% of LMPET fibers and 90 wt% of recycled high strength PET fibers, HP9K exhibits the optimal mechanical properties of those we tested in this study with 38.5 MPa of tensile strength, 1392.8 N/mm of tearing strength, 215.9 Kpa of bursting strength and 212.6 N of static stab resistance force. The combination of nonwoven and woven fabrics provides the benefits of their different stab behaviors, strengthening the puncture resistance of the hybrid-fabric fibrous planks. Most of all, an efficient recycling process and using textile and fiber waste to make protective fibrous planks decrease the production cost considerably, which makes the industrial and livelihood protective products more advantageous and acceptable.

In addition, the recycled high modulus PET fibers being generally coated with an oiling agent when produced prevents LMPET fibers from forming an adhesive layer during the hot pressing, and thus the interface bonding strength is low. Future studies need to remove the filling and oiling agent before conducting the test for further discussion.

## Figures and Tables

**Figure 1 polymers-11-01140-f001:**
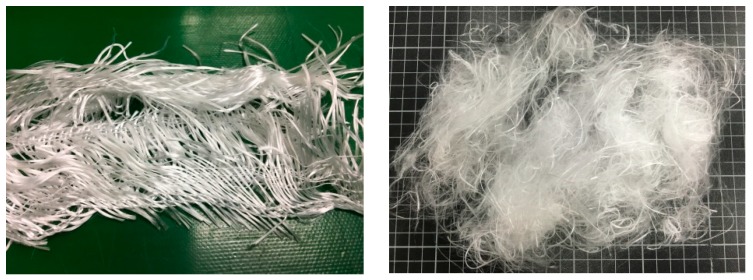
Images of recycled high strength PET selvages.

**Figure 2 polymers-11-01140-f002:**
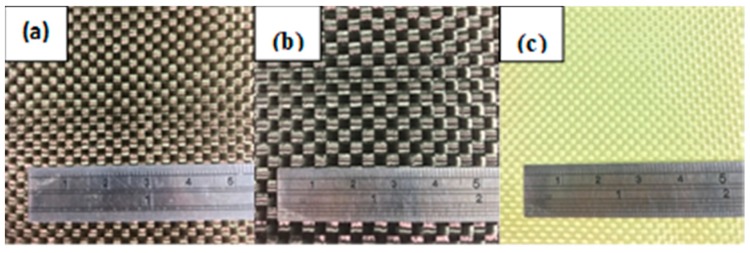
Images of (**a**) basalt; (**b**) carbon-fiber; and (**c**) aramid plain woven fabrics.

**Figure 3 polymers-11-01140-f003:**
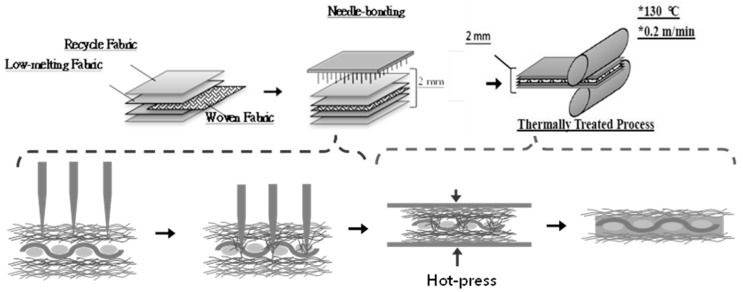
Manufacturing process of hybrid-fabric fibrous planks.

**Figure 4 polymers-11-01140-f004:**
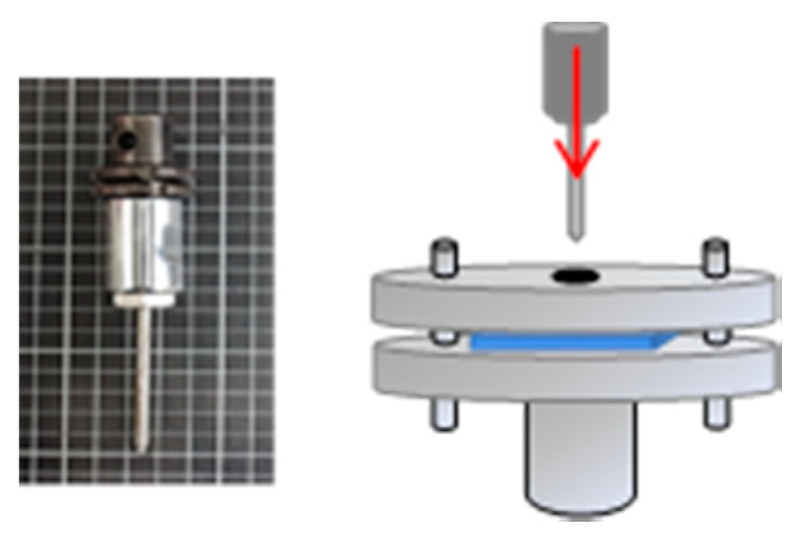
The equipment and puncture needle of static puncture resistance test.

**Figure 5 polymers-11-01140-f005:**
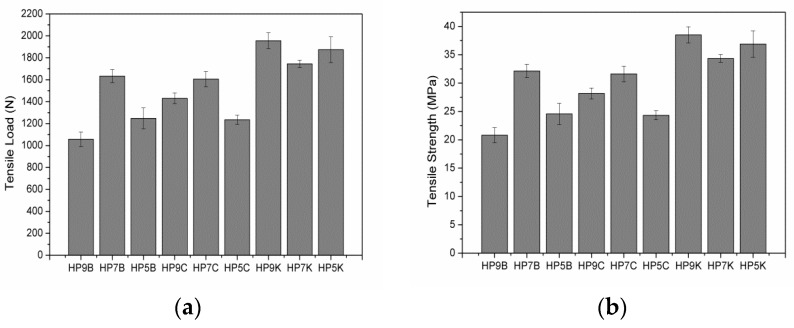
(**a**) tensile load and (**b**) tensile strength of hybrid-fabric fibrous planks as related to fiber blending ratios.

**Figure 6 polymers-11-01140-f006:**
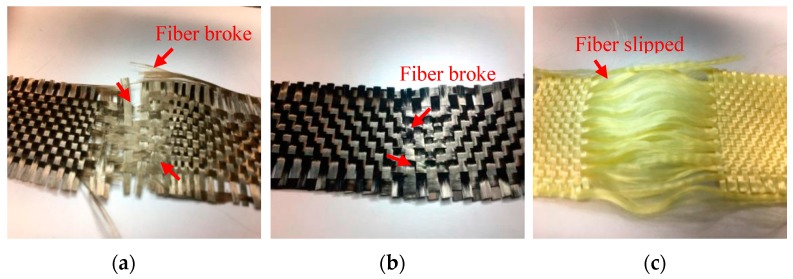
Damage level of (**a**) basalt; (**b**) carbon-fiber; and (**c**) aramid woven fabrics after tensile tests.

**Figure 7 polymers-11-01140-f007:**
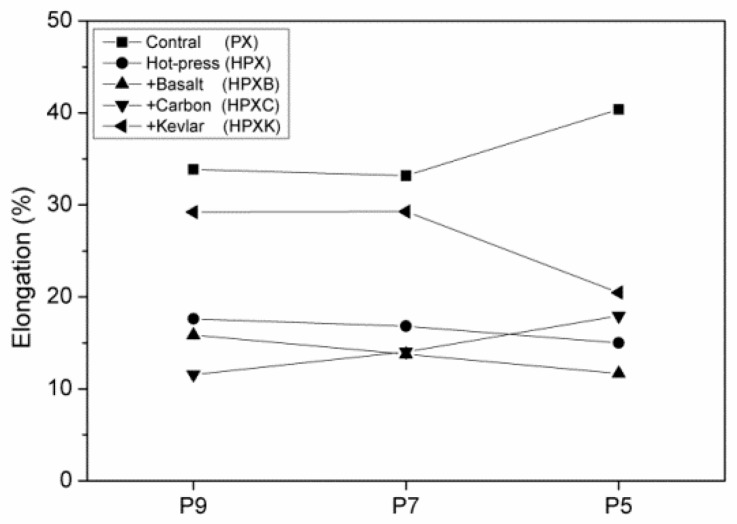
Elongation of hybrid-fabric fibrous planks as related to fiber blending ratios.

**Figure 8 polymers-11-01140-f008:**
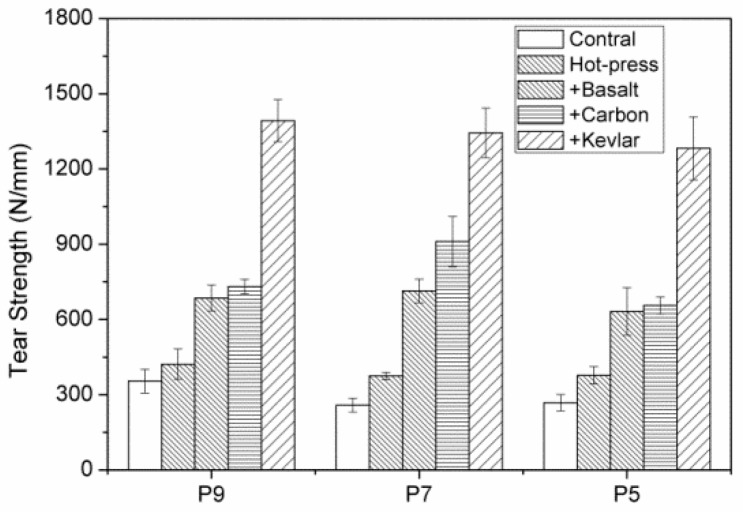
Tear strength of hybrid-fabric fibrous planks as related to fiber blending ratios.

**Figure 9 polymers-11-01140-f009:**
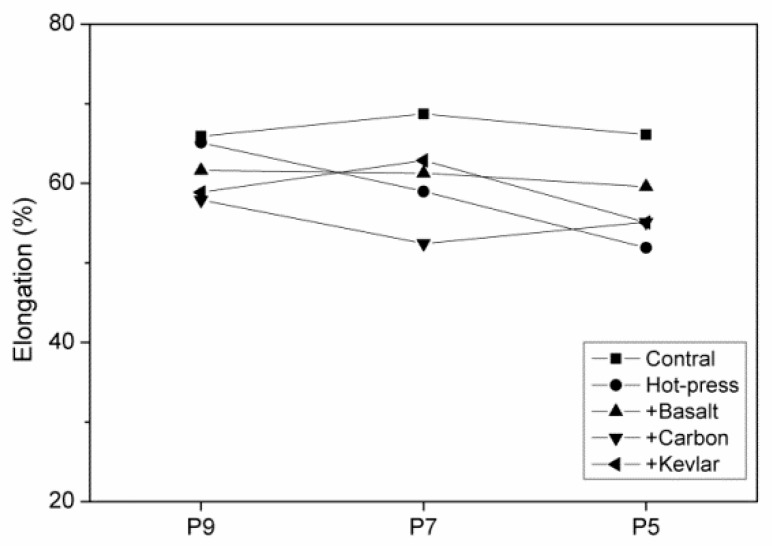
Elongation of hybrid-fabric fibrous planks as related to fiber blending ratios.

**Figure 10 polymers-11-01140-f010:**
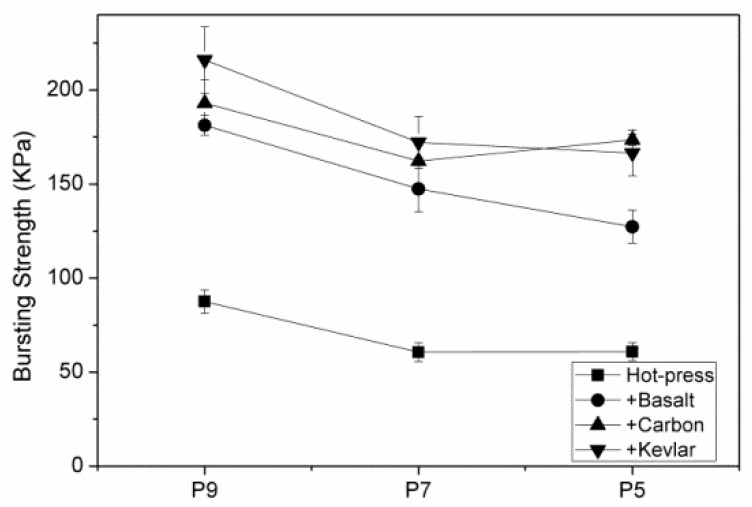
Bursting strength of hybrid-fabric fibrous planks as related to fiber blending ratios.

**Figure 11 polymers-11-01140-f011:**
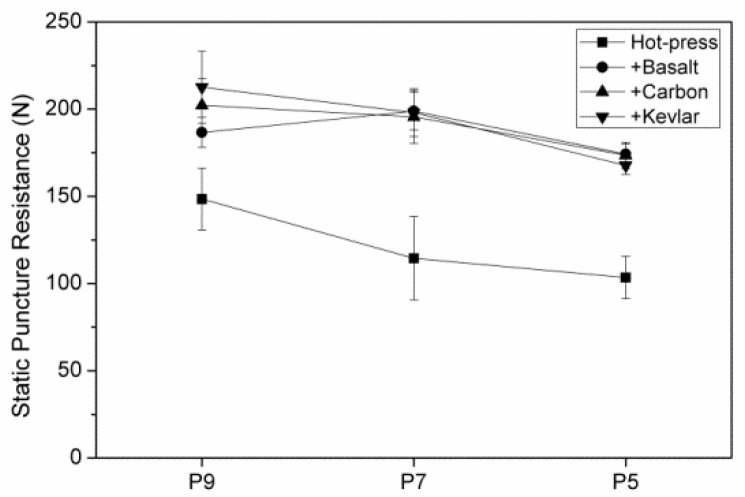
Static puncture resistance of hybrid-fabric fibrous planks as related to fiber blending ratios.

**Figure 12 polymers-11-01140-f012:**
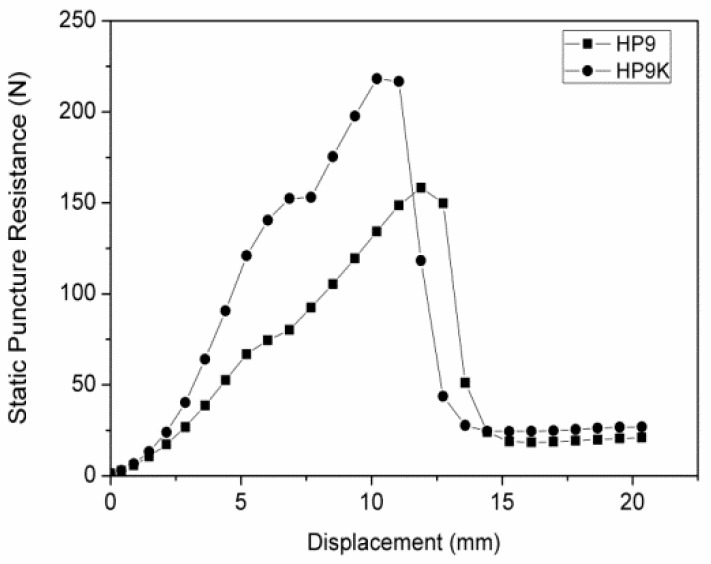
Static puncture resistance-displacement curve of hybrid-fabric fibrous planks of HP9 and HP9K.

**Table 1 polymers-11-01140-t001:** Physical properties of basalt, carbon fiber, and aramid plain woven fabrics.

Reinforced Woven Fabric	Fineness	Base Weight (g/m^2^)	Thickness (mm)	Tensile Load (N)
Basalt	2970 D	328	0.31	118.45
Carbon	12 K	390	0.60	164.63
Kevlar	1000 D	180	0.31	512.23

**Table 2 polymers-11-01140-t002:** Denotation and composition of hybrid-fabric fibrous planks.

Sample	RPET Content (wt%)	LPET Content (wt%)	Reinforcing Layer	Employment of Hot Press
P9	90	10	-	N
P7	70	30	-	N
P5	50	50	-	N
HP9	90	10	-	Y
HP7	70	30	-	Y
HP5	50	50	-	Y
HP9C	90	10	Carbon	Y
HP7C	70	30	Carbon	Y
HP5C	50	50	Carbon	Y
HP9B	90	10	Basalt	Y
HP7B	70	30	Basalt	Y
HP5B	50	50	Basalt	Y
HP9K	90	10	Kevlar	Y
HP7K	70	30	Kevlar	Y
HP5K	50	50	Kevlar	Y
LMPET Bonding Layer	-	100	-	-

**Table 3 polymers-11-01140-t003:** Physical properties of high strength PET matrices.

Experiment	RPET Content (wt%)	Tensile Strength, (MPa)	CV (%)	Elongation, (%)	Tearing Strength, (N/mm)	CV (%)	Elongation, (%)	Air Permeability, (cm^3^/cm^2^/s)
Without Hot-press	50 (P5)	13.3 ± 1.29	9.72	40.38 ± 1.36	267.9 ± 32.96	12.30	66.1 ± 7.13	49.3 ± 4.63
	70 (P7)	16.7 ± 1.92	11.48	33.17 ± 2.22	258.2 ± 27.63	10.70	68.7 ± 8.70	45.1 ± 3.75
	90 (P9)	16.0 ± 1.32	8.26	33.85 ± 2.40	354.4 ± 47.32	13.35	65.9 ± 9.42	40.1 ± 2.71
Hot-press	50 (P5)	13.8 ± 0.67	4.3	15.02 ± 1.04	376.7 ± 34.75	9.22	51.9 ± 7.09	28.4 ± 2.98
	70 (P7)	15.6 ± 0.80	5.79	16.82 ± 2.03	375.1 ± 14.68	3.91	59.0 ± 5.84	20.5 ± 3.53
	90 (P9)	17.0 ± 1.03	6.04	17.61 ± 1.79	422.0 ± 60.44	14.32	65.1 ± 2.58	15.8 ± 1.39

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
