# Peer review of "Mechanical and Static Stab Resistant Properties of Hybrid-Fabric Fibrous Planks: Manufacturing Process of Nonwoven Fabrics Made of Recycled Fibers"

_polymers, 2019, doi:10.3390/polym11071140_

Reviewer 1 Report

In this manuscript, the authors reported a decent work about the fabrication and evaluation of flexible fabric-based protective planks using recycled high strength PET fibers, which provided the fibrous planks with high tensile and tearing strengths. The combination of nonwoven and woven fabrics provided the benefits of their different stab behaviors, strengthening the puncture resistance of the hybrid-fabric fibrous planks. The manuscript is written well, and the characterization is comprehensive. However, a few detailed information is missing. It is only commended for publication after all the comments are addressed.

(1) What recycling method is used for the PET fibers used in this study? What’s the estimated cost?

(2) Please provide more information of the low-melting point polyester, such as the vendor, type, chemical structure, melting point, etc.

(3) What is the temperature for hot press?

(4) Needle-bonding is shown in Figure 3, but it is not introduced in the method section. It is also not mentioned in the results and discussion section. What is the purpose of the needle-bonding. Please explain and add this information.

(5) How many samples were tested for air permeability and mechanical properties? The mean and standard deviation value should be provided in Table 3.

Author Response

Reviewer1

Comments and Suggestions for Authors

In this manuscript, the authors reported a decent work about the fabrication and evaluation of flexible fabric-based protective planks using recycled high strength PET fibers, which provided the fibrous planks with high tensile and tearing strengths. The combination of nonwoven and woven fabrics provided the benefits of their different stab behaviors, strengthening the puncture resistance of the hybrid-fabric fibrous planks. The manuscript is written well, and the characterization is comprehensive. However, a few detailed information is missing. It is only commended for publication after all the comments are addressed.

(1) What recycling method is used for the PET fibers used in this study?

Thank you for the reviewer’s question.

The principal material in this study is recycled high strength PET selvages. In general, the staple fibers which used in non-woven fabrics are wavy or crimp in order to increase the friction. However, the waste selvages of woven fabric are usually cut from the edge of woven fabric made by the filament or continue yarn and still with woven fabric structure. So we have to break, dispersion, recycled and then reused these fibers. Therefore, the high strength PET waste selvages which are processed with the opening into recycled PET staple fibers.

What’s the estimated cost?

Thank you for the reviewer’s question.

It is well known that the non-woven process is simple in process, easy to process and high in efficiency, so its cost is low and the product is widely used. Therefore, in this study, the high strength PET waste selvages were reused through the non-woven process and made into Hybrid-Fabric. The cost is estimated to be only the purchase of waste selvages, the power consumption of the non-woven process machine and the loss of basic parts of the machine (for example Felting Needles, etc.).

(2) Please provide more information of the low-melting point polyester, such as the vendor, type, chemical structure, melting point, etc.

Thank you for the reviewer’s suggested.

We had revised and added more detail information in the Materials section.

Low-melting-point PET (LMPET)staple fibers (Far Eastern New Century, Taiwan) have a fineness of 4 D and a length of 51 mm, and are composed of a skin-core structure. The melting points of the skin and core are 110 °C and 265 °C.

(3) What is the temperature for hot press?

Thank you for the reviewer’s suggested.

We had revised and added more detail information in the Method section.

The sandwich-structured laminates are hot pressed into the hybrid-fabric fibrous planks which treated at 130 °C at a speed of 0.2 m/min and hot pressure with 10 MPa (two-wheel hot press machine, CW-NEB, Chiefwell Engineering Co., Ltd., New Taipei Citycity, Taiwan).

(4) Needle-bonding is shown in Figure 3, but it is not introduced in the method section. It is also not mentioned in the results and discussion section. What is the purpose of the needle-bonding. Please explain and add this information.

Thank you for the reviewer’s suggested.

We had revised and added more detail information in the Method section.

The PET staple fibers are mixed with low-melting point polyester (LMPET) fibers at ratios of 9:1, 7:3, and 5:5 to form high strength PET matrices by a needle punching machine (needle punching machine, SNP120SH6, Shoou Shyng Machinery Co., Ltd., New Taipei City, Taiwan) with a needle-punched speed of 200 needles/min and a line speed of 2.3 m/min. And during the needle-punching process, the needles are pressed in from the direction of the vertical fabric surface to laminate and bonded the multilayer web or multilayer fabric together.

(5) How many samples were tested for air permeability and mechanical properties? The mean and standard deviation value should be provided in Table 3.

Thank you for the reviewer’s suggested.

We had revised and added more detail data in table 3.

Table 3. Physical properties of high strength PET matrices

RPET content

(wt%)

Tensile Strength,

(MPa)

CV

(%)

Elongation,

(%)

Tearing Strength,

(N/mm)

CV

(%)

Elongation,

(%)

Air Permeability, (cm3/cm2/s)

Without Hot-press

50 (P5)

13.3±1.29

9.72

40.38±1.36

267.9±32.96

12.30

66.1±7.13

49.3±4.63

70 (P7)

16.7±1.92

11.48

33.17±2.22

258.2±27.63

10.70

68.7±8.70

45.1±3.75

90 (P9)

16.0±1.32

8.26

33.85±2.40

354.4±47.32

13.35

65.9±9.42

40.1±2.71

Hot-press

50 (P5)

13.8±0.67

4.3

15.02±1.04

376.7±34.75

9.22

51.9±7.09

28.4±2.98

70 (P7)

15.6±0.80

5.79

16.82±2.03

375.1±14.68

3.91

59.0±5.84

20.5±3.53

90 (P9)

17.0±1.03

6.04

17.61±1.79

422.0±60.44

14.32

65.1±2.58

15.8±1.39

Reviewer 2 Report

The authors of manuscript "Mechanical and Static Stab Resistant Properties of Hybrid-Fabric Fibrous Planks: Manufacturing Process of Nonwoven Fabrics Made of Recycled Fibers" have proposed to use waste fibers to produce strong protective fabric. Indeed, the manuscript is in right scope of the journal, but i see the length of the manuscript is too short to be considered in the journal of polymers.

Increase the content of the manuscript, for instance why the authors think there is a need of manufacturing of such protective fabric. Recycling aspect is given, but the authors used the commercially available recycled fibers. You can give some background as well.

Secondly, the results are presented but their explanation lacks literature backup. The conclusion part is too extensive, it can written to specify important findings in points.

Author Response

Reviewer2

Comments and Suggestions for Authors

The authors of manuscript "Mechanical and Static Stab Resistant Properties of Hybrid-Fabric Fibrous Planks: Manufacturing Process of Nonwoven Fabrics Made of Recycled Fibers" have proposed to use waste fibers to produce strong protective fabric. Indeed, the manuscript is in right scope of the journal, but i see the length of the manuscript is too short to be considered in the journal of polymers.

Increase the content of the manuscript, for instance why the authors think there is a need of manufacturing of such protective fabric.

Thank you for the reviewer’s question.

The amounts of selvages and textile wastes soar as a result of the manufacturers produce increasingly demanded protective textiles, including some high price, high strength, and high modulus fibrous materials. On the other hand, these protective products will be phased out after the end of their useful life. But the high modulus fibers used in these protective products may only be slightly damaged It will be a pity to just discard or burn these materials or use them as fillers, especially the materials that are left after the production of protective products. Therefore, in this study, we propose an efficient method to recycle the waste woven selvages and reused these fibers to develop a flexible stab resistance hybrid fabric composites which may be used in the protective clothing field and geotextiles field.

Recycling aspect is given, but the authors used the commercially available recycled fibers. You can give some background as well.

Thank you for the reviewer’s suggested.

We had revised and added more detail information in the introduction section.

In the production of weaving fabrics, due to the limitations of the machine and the quality of the product, the width of the fabric usually produced must be slightly larger than the actual required width. Therefore, almost all fabrics need to be cut and then sold. A large amount of discarded cloth selvages are produced in the process.

The principal material in this study is recycled high strength PET selvages. In general, the staple fibers which used in non-woven fabrics are wavy or crimp in order to increase the friction. However, the waste selvages of woven fabric are usually cut from the edge of woven fabric made by the filament or continue yarn and still with woven fabric structure. So we have to break, dispersion, recycled and then reused these fibers. Therefore, the high strength PET waste selvages, which are processed with the opening into recycled PET staple fibers.

Secondly, the results are presented but their explanation lacks literature backup.

Thank you for the reviewer’s suggested.

We had revised and added more citied reference to be more convincing in the result and discussion section.

22.         Correia, N.; Bueno, B. Effect of bituminous impregnation on nonwoven geotextiles tensile and permeability properties. Geotextiles and Geomembranes 2011, 29, 92-101.

23.         Lin, M.C.; Lou, C.W.; Lin, J.Y.; Lin, T.A.; Lin, J.H. Mechanical property evaluations of flexible laminated composites reinforced by high-performance Kevlar filaments: Tensile strength, peel load, and static puncture resistance. Composites Part B 2019, 166, 139-147.

24.         Zakriya, G.; Ramakrishnan, G. Insulation and mechanical properties of jute and hollow conjugatedpolyester reinforced nonwoven composite. Energy and Buildings 2018, 158, 1544-1552.

25.         Rawal, A.; Saraswat, H. Stabilisation of soil using hybrid needlepunched nonwoven geotextiles. Geotextiles and Geomembranes 2011, 29,197-200.

The conclusion part is too extensive, it can written to specify important findings in points.

Thank you for the reviewer’s suggested.

We had revised the conclusions section, and added more describe as below:

This study proposes flexible fabric-based protective planks, which are recycled high strength PET fibers by processed with minimum damage for secondary production, thereby obtaining recycled high performance fibers with relatively lower production cost. In this study, the different reinforcing woven fabrics are combined with matrices to form hybrid-fabric fibrous planks. Despite multiple combining and carding processes, the recycled PET staple fibers are proven to provide the fibrous planks with high tensile and tearing strengths. The test results indicate that recycled PET fibers remains the high strength and can be made into protective products.

The test results indicate that recycled PET fibers remains high strength and can be made into protective products. The sample consisting of 10 wt% of LMPET fibers and 90 wt% of recycled high strength PET fibers, HP9K exhibits the optimal mechanical properties those we tested in this study with 38.5 MPa of tensile strength, 1392.8 N/mm of tearing strength, 215.9 Kpa of bursting strength and 212.6 N of static stab resistance force. The combination of nonwoven and woven fabrics provides the benefits of their different stab behaviors, strengthening the puncture resistance of the hybrid-fabric fibrous planks. Most of all, an efficient recycling process andusing textile and fiber waste to make protective fibrous planks decreases the production cost considerably, which makes the industrial and livelihood protective products more advantageous and acceptable.

In addition, due to the recycled high modulus PET fibers are generally coated with oiling agent when it produced, which prevents LMPET fibers to form an adhesive layer during the hot pressing, and thus the interface bonding strength is low. The future studies need to remove the filling and oiling agent before conducting the test for further discussion.

Round  2

Reviewer 1 Report

The authors addressed all my questions. The manuscript is recommended to publish.

Author Response

Reviewer 1
Comments and Suggestions for Authors

The authors addressed all my questions. The manuscript is recommended to publish.

Thanks again to the comments of the reviewers.

Reviewer 2 Report

The authors have answered all the comments, however, i still feel the manuscript lacks literature review. The authors can add these two reference to emphasize non-wovens. 1. (Hussain et al. LMPP Effects on Morphology, Crystallization, Thermal and Mechanical Properties of iPP/LMPP Blend Fibres. Fibres & Textiles in Eastern Europe.

2. (Hussain et al. Optimization of mechanical and thermal properties of iPP and LMPP blend fibres by surface response methodology. Polymers.), and this for recycling of textiles, 3. (Yasin et al. Global consumption of flame retardants and related environmental concerns: A study on possible mechanical recycling of flame retardant textiles. Fibers).

Author Response

Reviewer2
Comments and Suggestions for Authors

The authors have answered all the comments, however, i still feel the manuscript lacks literature review. The authors can add these two reference to emphasize non-wovens. 1. (Hussain et al. LMPP Effects on Morphology, Crystallization, Thermal and Mechanical Properties of iPP/LMPP Blend Fibres. Fibres & Textiles in Eastern Europe.

2. (Hussain et al. Optimization of mechanical and thermal properties of iPP and LMPP blend fibres by surface response methodology. Polymers.), and this for recycling of textiles, 3. (Yasin et al. Global consumption of flame retardants and related environmental concerns: A study on possible mechanical recycling of flame retardant textiles. Fibers).

Thanks again to the comments of the reviewers.

We had revised and added these citied references in this text.